# The Tumor Microenvironment in Liver Metastases from Colorectal Carcinoma in the Context of the Histologic Growth Patterns

**DOI:** 10.3390/ijms22041544

**Published:** 2021-02-03

**Authors:** Gemma Garcia-Vicién, Artur Mezheyeuski, María Bañuls, Núria Ruiz-Roig, David G. Molleví

**Affiliations:** 1Tumoral and Stromal Chemoresistance Group, Molecular Mechanisms and Experimental Therapy in Oncology Program (ONCOBELL), Institut d’Investigació Biomèdica de Bellvitge—IDIBELL, 08908 L’Hospitalet de Llobregat, Spain; ggarciav@idibell.cat (G.G.-V.); maria.banuls@iconcologia.net (M.B.); nuria.ruiz@bellvitgehospital.cat (N.R.-R.); 2Program Against Cancer Therapeutic Resistance (ProCURE), Catalan Institute of Oncology, 08908 L’Hospitalet de Llobregat, Spain; 3Department of Immunology, Genetics and Pathology, Uppsala University, 752 37 Uppsala, Sweden; artur.mezheyeuski@igp.uu.se; 4Department of Pathology, Hospital Universitari de Bellvitge, 08908 L’Hospitalet de Llobregat, Spain

**Keywords:** histologic growth patterns, desmoplasia, carcinoma-associated fibroblasts, microenvironment, liver, hepatic stellate cells, capsule

## Abstract

Colorectal carcinoma (CRC) is the third most common cancer. Likewise, it is a disease that has a long survival if it is prematurely detected. However, more than 50% of patients will develop metastases, mainly in the liver (LM-CRC), throughout the evolution of their disease, which accounts for most CRC-related deaths. Treatment it is certainly a controversial issue, since it has not been shown to increase overall survival in the adjuvant setting, although it does improve disease free survival (DFS). Moreover, current chemotherapy combinations are administered based on data extrapolated from primary tumors (PT), not considering that LM-CRC present a very particular tumor microenvironment that can radically condition the effectiveness of treatments designed for a PT. The liver has a particular histology and microenvironment that can determine tumor growth and response to treatments: double blood supply, vascularization through fenestrated sinusoids and the presence of different mesenchymal cell types, among other particularities. Likewise, the liver presents a peculiar immune response against tumor cells, a fact that correlates with the poor response to immunotherapy. All these aspects will be addressed in this review, putting them in the context of the histological growth patterns of LM-CRC, a particular pathologic feature with both prognostic and predictive repercussions.

## 1. Introduction

Colorectal carcinoma (CRC) is the third most frequent tumor in the world and the second in Europe. Taking both sexes into account, there are 1,850,000 new cases annually worldwide [1]. The incidence of this tumor increases by 2% each year and it is the primary cause of death due to cancer. The annual number of deaths is approximately half that of the incidence (880,000 deaths worldwide in 2020). Therefore, tackling this disease is one of the highest priority challenges in healthcare. Despite the best survival results with adjuvant chemotherapy since the publication of the MOSAIC study [2], approximately 20% of patients will develop metachronous metastases in the liver in less than three years following resection of the primary tumor and 25% displayed liver metastases at diagnosis (synchronous) [3]. In liver metastases from CRC (LM-CRC), curative surgery is the treatment of choice if it is performed in specialized centers. However, in these advanced stages of the disease, the 5-year survival rates are less than 10%. Although only about 15% of metastases are initially resectable, this percentage can reach 30% if conversion chemotherapy is administered to selected patients. In the past few years, many treatment advances, such as more effective chemotherapy or staged hepatectomies, have emerged. However, despite the curative intent, there is a high rate of intrahepatic recurrence. Moreover, despite the fact that many retrospective studies have identified poor prognostic factors such as tumor size or number of lesions [4], none of them provide enough information to take surgical decisions. This fact highlights the need for new prognostic factors to select the best therapeutic approach for each patient. 

In addition, a few advances in preclinical research have been translated into significant benefits, either in clinical terms or with respect to the quality of life of patients with LM-CRC. Despite important preclinical discoveries about aberrant biological pathways associated with the development and progression of the primary tumor, certain pathological aspects of the formation of liver metastases remain unknown.

A very common approach to study the biology of the metastatic process has been to compare the primary tumor and the metastatic lesions. Although great similarities between both tumors have been described at the genomic level [5], at the transcriptional level, relevant differences that reflect the particular microenvironment of each different location have been determined. The liver, to name a few examples, has exclusive stromal cells at this location, e.g., Kupffer cells, hepatic stellate cells (HSC) and portal fibroblasts (PF). Due to the great influence of the microenvironment on both the tumor biology and the response mechanisms to chemotherapy and targeted therapies, this fact should have a determining impact on the clinical management of patients with LM-CRC compared to patients without distant lesions. 

Moreover, LM-CRC present different histological characteristics that accentuate their morphological differences from primary colorectal tumors. Thus, a remarkable presence of cancer-associated fibroblasts (CAFs), constituting the main stromal cell type and reaching 75–80% of the entire tumor mass, high interstitial pressure that hampers the delivery of drugs and a defective immunologic response from the host are essential to understand the therapeutic resistances of LM-CRC. Even considering possible therapeutic strategies focused on stromal cells, we must bear in mind that fibroblasts demonstrate certain topographic differentiation [6]. This means that the CAFs of a primary tumor may present different characteristics to the CAFs of a liver lesion [7,8] in the same patient, and therefore therapies designed against colonic CAFs may not be efficient enough over hepatic CAFs. On the other hand, liver metastases have displayed a unique histologic feature, a histologic growth pattern (HGP) [9], that will be described in more detail in the next section (Figure 1). Interestingly, these HGPs displayed prognostic [10,11] and predictive [12] value, suggesting that possibly intratumoral CAFs and peritumoral CAFs are exerting different functions that could be a consequence of either different origins or different cellular precursors, or particular responses of the host against the tumor type.

Thus, the tumor microenvironment in liver lesions presents unique and differential characteristics that are going to be highly relevant in the management of the patients. In the following sections, we will elucidate the relevance of the different cell types that shape up the liver microenvironment, the way in which they are structured, the altered immune response and the clinical and therapeutic consequences that can be derived from them.

## 2. Histology of LM-CRC

As mentioned before, LM-CRC are characterized at the histological level by their way of growing and infiltrating liver parenchyma. It is widely accepted by the scientific community that there are three main HGPs named “pushing”, “replacement” and “desmoplastic”, which were described for the first time in 1996 (including a fourth subtype, sinusoidal) [13], although the terminology was finally coined in 2001 [14] (Figure 1). The “desmoplastic” HGP is also named “encapsulated” and its most important feature is that tumoral cells are separated from liver parenchyma by a fibrotic stromal capsule; thus, tumoral cells and hepatocytes never touch. Conversely, in the “replacement” subtype, there is a direct contact between hepatocytes and tumoral cells. In “replacement” HGP, also called “invasive”, tumor cells infiltrate liver parenchyma, replacing the hepatocytes and following the hepatic sinusoidal structure, which makes it difficult to clearly distinguish the tumor border [15]. Finally, in the “pushing” or “expansive” HGP, tumoral cells literally “push” the hepatocytes, which become flattened in consequence, and the tumor border is clearly demarcated, but no fibrotic tissue intervenes. However, there is no direct contact between hepatocytes and tumor cells. 

The relevance of these distinct HGPs falls in their possible utility as a prognostic and predictive tool, due to its numerously described association with tumor recurrence and overall survival [10,16,17,18] as well as prediction of response to anti-angiogenic therapies [12]. On this matter, patients who present a predominantly desmoplastic growth pattern have superior survival after resection with curative intent when compared with patients presenting the replacement type [9,10,11,16,18,19]. Although a significant percentage of patients present a mixed pattern, with dHGP and non-dHGP areas, the most favorable prognosis is in those patients who present a pure dHGP pattern, complete throughout the entire liver–tumor interface [10]. Interesting, prognosis has also been related to the thickness of the capsule: the thicker the capsule, the better the prognosis [20].

These differences in survival could partially rely on the fact that, in the desmoplastic growth pattern, there is a band of stroma enriched in collagen and with dense lymphocytic infiltration that may act as a physical obstacle for the tumor expansion. This is particularity emphasized by an increase in collagen type IV and integrin blockade, which reduces the infiltration capacity of tumoral cells through liver parenchyma [14,18]. 

In this line, the differences in recurrence rates could be explained by the HGP surgical implications. For mCRC, liver resection is the gold standard therapy, with a curative aim. The success of this therapy is closely related to the percentage of R0 resections achieved. It has been described that desmoplastic patterns are associated with an increased rate of R0 (tumor-cell-clean resection margins), probably because the stromal tissue surrounding the metastases protects against a margin-positive resection. Therefore, a resection with a more extended margin could benefit replacement and pushing cases [21]. 

Interestingly, HGPs have also been described in liver metastases from other types of cancer such as uveal melanoma [22,23], gastric cancer [24] and breast cancer [25], which adds interest to the predictive value of HGPs. However, because hepatectomy is less common in these cases, the prognostic role of HGPs is much less explored [21].

The lung is the second most common place where CRC metastasizes [26]. In fact, three different HGPs have also been described in CRC lung metastases. First, a pushing pattern, where tumoral cells compress pulmonary parenchyma. Second, a desmoplastic pattern, which also shows a desmoplastic rim containing immune infiltrates. Finally, third, an aerogenous pattern that can be considered as the analogue to the replacement pattern in liver as the tumor cells spread in the pulmonary air spaces without disrupting the lung architecture [27]. These HGPs observed in lung metastases also have clinical meaning, and the aerogenous pattern displays a poor prognosis, which concurs with liver replacement HGP as well. The brain is another organ where three metastatic patterns have been reported: a well-demarcated one, vascular co-option and diffuse infiltration; however, their impact on overall survival could not be assessed in autopsy studies [28]. Therefore, HGPs could have a role in guiding therapeutic decisions in many different contexts apart from LM-CRC. This also implies the need for predicting the HGP on different imaging modalities.

Another aspect to consider is how preoperative neoadjuvant treatment received by a proportion of patients with LM-CRC could affect the HGP and the ratios of cells that constitute the metastatic tumor microenvironment (TME). Concerning this, it has been described that in neoadjuvantly treated patients, the percentage of dHGP is higher. However, further investigation is needed, as most of the studies that attend this issue englobe different chemotherapeutic regimens [10]. Moreover, it has been reported that some features, such as the expression of a plasminogen receptor or the presence of CD68+ macrophages, differ between desmoplastic and non-desmoplastic HGPs but not in neoadjuvantly treated patients [29]. Thus, neoadjuvant chemotherapy is altering tumor growth in many different aspects, the study of which could be crucial for the development of better treatment strategies.

Taken together, HGPs are not only a distinct morphological feature of LM-CRC, but also a valuable prognostic marker. HGPs also reflect the biological processes underlying tumor growth. Moreover, therapy-induced conversion of non-dHGP into dHGP would improve the survival for the patients with metastatic disease dramatically. The development of such treatment, however, requires better understanding of the biological processes underlying metastatic growth.

## 3. Biology of LM-CRC

From a biology perspective, the mode through which metastases grow in the liver has awaked high interest as it should conceal the mechanisms underlying the previously described differences in prognosis. First, it is important to take into account that the progression of liver metastases consists of four phases, proposed for the first time by Vidal-Vanaclocha in 2011 [30], in which the hepatic microenvironment can play opposing roles [31]. The first one, called the microvascular phase, occurs in sinusoidal vessels when circulating tumor cells arrive [31,32]. At this timepoint, cancer cells need to be able to attach and adhere to the endothelial cell layer and, eventually, transmigrate through the vascular endothelium. This complex process is called extravasation and leads to the following second phase: pre-angiogenic or intra-lobular micrometastatic phase. In this phase, stromal cells from liver and immune cells are recruited, but the micrometastases still do not show vascularization. In the third angiogenic or pan-lobular phase, the hypoxic microenvironment induces the recruitment of endothelial cells and formation of blood vessels. Finally, in the fourth lobar growth phase, the new tumor can be clinically detectable. The progression of these phases and the following growth of the metastasis depend on the interactions that occur between malignant cells and the liver microenvironment [31,32]. 

Thus, this process can be assessed in two ways. The local microenvironment can be considered as the major determinant of the host reaction on the tumor progression. In this case, the host tissue defines the development of a certain HGP and thus contains the key for explaining worse or better outcome. On the other hand, metastasized cancer cells interact with the different host cell types and, through the paracrine signaling, transform host stroma and determine HGP development. However, up to now, the formation of a particular HGP is still an unknown process, and there are no clues for a tumor-driven or a host-driven mechanism. A concept proposed by Fernando Vidal-Vanaclocha [33] distinguishes at least three different “entry points” for the cancer cells and thus three different metastatic niches which shape up the tumor–host interaction and drive the evolution of the tumor: periportal, perisinusoidal and intrasinusoidal. According to this concept, cancer cells which enter periportal and perisinusoidal spaces develop according to the “classical” process, following four phases, as described above. During the intra-lobular and pan-lobular phases, malignant cells invade parenchymal cell plates (perisinusoidal location) or the portal space (periportal location) and develop a pushing–desmoplastic phenotype, while intrasinusoidal metastases start proliferation already in the intravascular space, avoiding the extravasation phase. These metastases develop in a very distinct microenvironment, surrounded by liver sinusoidal endothelial cells (LSEC) and elements of the blood, which require different surviving mechanisms. During further growth, intrasinusoidal micrometastasis disrupts the sinusoid and develops a replacement HGP. However, these processes have not been experimentally demonstrated so far.

Concerning the host microenvironment, the liver harbors different stromal cells, which include host macrophages (Kupffer cells), host fibroblasts (HSC and PF), LSEC, liver-associated lymphocytes and dendritic cells [31,32]. Moreover, there are also circulating and bone marrow-derived immune cells that are recruited in response to the malignant growth. These host cells, as well as the recruited cells of the immune system, interact in a complex interplay of cytokines and chemokines that conform to the microenvironment in which the metastatic tumor growth takes place.

### 3.1. Liver Sinusoidal Endothelial Cells (LSEC)

LSEC play a crucial role, as they are the first host cells to be in direct contact with tumoral cells. LSEC are part of the first line of defense, also accompanied by Kupffer cells and hepatic natural killers, which may destroy tumor cells through the release of pro-apoptotic signals prior to extravasation [31,34]. However, LSEC can also promote attachment and adherence of the cancer cells by expressing cell surface adhesion molecules (CAM), thus facilitating metastasis formation [31]. Moreover, LSEC secrete fibronectin, which can induce an epithelial–mesenchymal transition in cancer cells, and macrophage migration inhibitory factor [35,36]. In addition, LSEC can also be a key player for HGP development as they are closely related to the metastasis vascularization pattern [31,37,38,39].

Finally, LSEC are endocytic cells that scavenge molecules from the bloodstream and present them to hepatic lymphocytes, which leads to a physiological immune regulation towards tolerance preventing liver parenchymal damage. However, during hepatic metastasis development, tumor-activated LSEC become highly inflamed which fosters their immune suppressive effect, especially in the replacement HGP [27]. As shown in an experimental model of hepatic CRC metastasis, the mechanism is contributed by IL-1-induced LSEC mannose receptors [40].

### 3.2. Cancer-Associated Fibroblasts

As it happens in primary tumors [8], LM-CRC are characterized by desmoplasia, which is defined as the presence of a high amount of stroma among the malignant cells. This reactive stroma creates the proper environment for tumor development and CAFs are the main component of it. In a desmoplastic TME, CAFs interact with the extracellular matrix and with the other cells and mediate processes that favor tumor cells: promoting tumor cell proliferation and migration, supporting stem cell niche generation, regulating immune suppression and influencing chemoresistance.

In general terms, fibroblasts in physiological conditions remain in a quiescent state and become activated in case of tissue damage. Similarly, CAFs are fibroblasts that have been activated by the interaction with tumor cells. This activation implies morphological changes and the expression of a plethora of CAFs markers, such as α-smooth muscle actin, periostin and fibroblast activating protein (FAP). This marker expression pattern changes markedly depending on the CAFs’ location both inside the tumor and in their anatomic demarcation [7]. Once they are activated, CAFs generate the collagens and fibronectin that conform to the extracellular matrix, as well as crosstalking with tumor cells through cytokines and growth factors [41]. 

Moreover, in different types of cancer, the quantity of CAFs is associated with bad prognosis [42,43,44]. However, the existence of different CAF subpopulations associated with different functions has been demonstrated [45,46]. Thus, they have to be considered as heterogeneous cells and the biomarkers to characterize these different subpopulations need to be defined [8,47,48]. 

In addition, CAFs also play an important role in the process of chemoresistance [49]. They can create a physical barrier that hinders the drugs to reach the tumor cells or alter drugs’ function through factors production [50,51,52].

For all these reasons, CAFs are considered to be a potential therapeutic target for desmoplastic tumors. Therefore, many research groups have focused on developing therapeutic strategies for the elimination of CAFs, which have shown opposite effects. Thus, the inhibition of CAFs’ functions had a positive effect on sensitizing tumoral cells to treatments [48,53,54,55]. On the contrary, other studies have shown that CAF depletion can also result in a protumoral effect [53]. Therefore, this further suggests the coexistence of different CAF subpopulations associated with different functions [53].

In the context of LM-CRC, the function of CAFs remains unclear. However, as these metastases have been shown to grow in different HGPs associated with prognosis, CAFs located in particular topographic areas of the tumor, e.g., peritumoral/encapsulating CAFs, CAFs in invasive margins or CAFs in central tumoral areas, might be playing different roles and probably reflect different temporal processes (Figure 2).

In other types of cancer, such as pancreas [56], breast [57] and lung cancer [58,59], different subpopulations of CAFs have already been described. These publications highlight the fact that CAFs may change from one state to another depending on the context [60]. Other authors have classified CAFs according to their spatial location in pancreatic tumors, having observed a particular transcriptional signature at each location [61]. Despite these results, clearly more research is needed to elucidate the most accurate classification and association of particular subsets with functionality and spatial location.

Taking together, these data provide evidence that there may coexist subpopulations of CAFs performing both tumor-supportive and tumor-suppressive functions. These subpopulations can be shaped up by multiple factors such as the location inside the tumor, the origin of the CAF and the effect of chemotherapy and may vary between different tissues and anatomic demarcations. In this line, characterizing the degree of fibroblast heterogeneity as well as the dynamics of the different biomarkers would provide highly valuable tools to design new therapeutic strategies. 

Concerning the stromal reaction of the three HGPs, they can be simplified in two groups, the ones which show a fibrotic rim, the desmoplastic HGP (dHGP), and the ones that do not (non-dHGP). In this context, three different topographic locations of CAFs can be distinguished. First, CAFs present in the fibrotic rim of the dHGP, i.e., in peritumoral stromal areas. Second, CAFs located at the invasive margin, i.e., intratumoral CAFs located in outer tumoral areas. As the prognoses differ drastically in metastases that show 100% of dHGP, it may be plausible that peritumoral and intratumoral fibroblasts near to the host–tumor interface are different subsets with different functions, or particular juxtacrine and paracrine crosstalk with tumor cells might induce phenotypic differences with functional consequences (Figure 2). Third, intratumoral CAFs located in central tumoral regions, common for the different HGPs. These areas usually present large extensions formed mainly by myofibroblastic CAFs and extracellular matrix components and are often related to secondary fibrosis processes. These are areas where CAFs do not usually express FAP or other inflammatory CAF markers. In colloquial terms, we could say that these are areas where the battle between the tumor and the host has already ended—areas of injury that must be repaired. 

The fibrotic rim of the dHGP is considered by many authors as a response to the inflammation and damage in the host tissue to the growing tumor. Moreover, the link between the presence of the desmoplastic capsule and a considerable better prognosis makes it possible to hypothesize that these CAFs play a tumor suppressor role, limiting physically and chemically the tumor growth and attracting immune infiltrates to act against malignant cells. 

A few results seem to indicate that the fibrotic rim could be acting as a defensive barrier. First, the capsule could be mechanically acting as a protective shield avoiding local spreading. Second, it might be acting as a chemical barrier. Lunevicius et al. [62] described that the inner side of the capsule was enriched in collagen fibers and fibroblastic cells negative for metalloproteases. In contrast, the outer portion of the capsule was formed by cells expressing matrix metalloproteins (MMPs). In a similar vein, Illemann et al. [63] also described a difference in the cellular composition of the capsules, with the outer portion being enriched in αSMA-positive CAFs. What we observed in our laboratory is a certain gradient of positivity for some characteristic CAF markers. As summarized in Figure 3, we observed that the outer part of the capsule, the half side in contact with hepatocytes, is enriched in αSMA-expressing cells, while the inner part, in contact with tumor cells, is enriched in FAP-positive CAFs. 

This gradient was also observed for periostin, fibronectin and podoplanin, which indicates that a certain differential degree of activation or a differential cellular composition could be occurring (Figure 4).

The differences observed in terms of the differential expression of certain proteins in the fibrotic rim raises the controversy whether this fact is due to the presence of different types of CAFs depending on their location in the capsule. In turn, the coexistence of different CAFs could imply that each of them came from a different precursor, a fact that has yet to be demonstrated. Another possibility is that it is a single cell type with high plasticity, as a consequence of the influence of tumor cells on the inner half of the capsule. However, defining the origin of activated myofibroblasts is one of the major challenges of the research on fibrosis, as those cells might be potential therapeutic targets against this remodeling process. In addition, several fibroblastic cells have been defined in the human liver [64]. In animal models, many different fibroblast origins have been also identified. Fibrocyte invasion, endogenous resident fibroblasts, pericytes and gli1-positive mesenchymal stem cells would be some examples. In the context of cancer, it is also a recurrent aspect to focus on, especially in the desmoplastic kind of tumor. This is the case of pancreatic cancer, where the two major stromal sources are resident pancreatic fibroblasts and stellate cells. In breast cancer, CAFs are thought to derive from pericytes, resident fibroblasts and even from the transformed epithelial cells via EMT. Thereby, the origin of CAFs, as well as the origin of myofibroblasts in fibrosis, varies depending on the tissue [57,65].

In the context of LM-CRC and fibrotic diseases of the liver, the three main sources described are resident mesenchymal cells, which include quiescent HSC and resident PF, but we cannot exclude other cells that have undergone EMT (hepatocytes, cholangiocytes and endothelial cells), and bone-marrow derived cells, consisting of fibrocytes and circulating mesenchymal cells. However, the origin of myofibroblasts may be different depending on the liver disease that causes the fibrosis [66].

In LM-CRC, many cell types have been discussed to contribute to the stromal reaction, such as smooth muscle cells or circulating mesenchymal cells. However, as currently considered, the main CAF sources in the liver are HSC, PF [67] and fibroblasts surrounding the centrolobular vein, the so-called second-layer cells [68].

HSC are in the space of Disse, around the sinusoidal LSEC, and represent 5% to 8% of the liver cellular population. In normal conditions, they are characterized for being in a quiescent state and containing lipid droplets with vitamin A [68]. Once activated, HSC acquire myofibroblast features such as αSMA expression and extracellular matrix deposition and become the orchestrating cells of the liver injury response [69]. In vitro, these cells have been reported to be able to undergo myofibroblastic transdifferentiation [70]. In hepatic metastases, HSC release growth factors and metalloproteinases and deposit type I and IV collagens and laminin [31]. They are also involved in endothelial cell recruitment and neoangiogenesis [71], as well as in tumor cell invasion and proliferation [31]. On the other hand, PF can also be the origin of CAFs when cancer cells are located in portal tracts, but they probably initiate the stromal response through a different process. 

Finally, another origin that has been explored are hepatocytes located at the periphery of the metastases, which can undergo EMT and contribute to fibrosis and angiogenesis [70].

In the context of HGP and spatial localization, the different types of CAFs lead us to question which cell type is the precursor of fibrotic capsules or if is it a mixture of the cell types described above. In this sense, we can learn a lot from the research carried out on fibrotic liver diseases, which will probably follow processes similar to liver tumoral malignancies. Therefore, it is crucial to know in detail the differences between HSC and PF, especially in the context of the activation of both cell types. This can also be affected by the route of entrance of the tumor cell into the liver, that is, whether it extravasates in a centrolobular vein, in a portal tract or directly into a sinusoid. It has been observed in experimental models of fibrosis that depending on the etiology, there is a differential activation of both HSC and PF. For example, HSC are the main source of myofibroblasts in lesions induced by Tetrachloromethane (CCl_4_), whereas PF are the main source in lesions induced by bile duct occlusion [72]. Likewise, the differences between activated HSC and activated PF have been characterized, and some characteristic biomarkers of each cell type have been reported once they have undergone an activation process. It has been described that PF are characterized by the expression of CD90, while activated HSC would be negative for this marker [73]. On the contrary, cytoglobin has been described as a good characteristic marker of activated HSC and that it is not expressed in activated PF [74]. What seems very plausible is that CAFs from different origins coexist in LM-CRC. Data from our laboratory reinforce this fact, since we isolated CD90-positive CAFs and CD90-negative CAFs that could correspond to a differential origin or precursor (data not shown).

However, we cannot rule out that the generation of the capsule is due to intrinsic properties of the tumor cell. In any case, they are important aspects that must be taken into account because they could provide future therapeutic strategies.

Therefore, all this knowledge leads us to speculate that each HGP contains CAFs coming from a distinct origin among those earlier described. This would be in line with the hypothesis on the HGP predetermination by the different access routes of circulating tumor cells into hepatic tissue [30]. On the one hand, if tumoral cells enter through a portal tract, they would first interact with PF and those would mediate the stromal reaction. On the other hand, if they reach hepatic sinusoids, they will be able to activate HSC and the process of fibrosis would be orchestrated by them [75,76].

### 3.3. Tumor Vascularization

Different HGPs have also been associated with different types of tumor vascularization. In fact, another common classification for HGPs considers only two patterns: the angiogenic, which includes desmoplastic and pushing HGPs, and the non-angiogenic one that refers to the replacement HGP [27]. 

In the replacement HGP, the tumoral cells benefit from the hepatic architecture: they progressively “replace” the hepatocyte plates and use the already existing connective tissue and blood vessels. Hence, tumor vascularization is carried out by a non-angiogenic process called vessel “co-option” [12,14,25]. On the other hand, in desmoplastic and pushing HGPs, the tumor completely disturbs the hepatic architecture, creating its own supporting stroma, new blood vessels through an angiogenic process and, in the dHGP, forming the desmoplastic rim that separates the tumor from parenchyma. As an effect of vascular endothelial growth factor (VEGF), these new angiogenic blood vessels are composed by proliferating endothelial cells and form what is called “vascular hot spots”. Moreover, as they are not covered by pericytes, VEGF also induces the deposition of fibrin in a perivascular space [16,25].

Some studies have suggested that these different tumor vascularization types can be related to the route that tumoral cells take during the extravasation phase. In this line, a study performed with a murine model found that, if the cancer cells had invaded between LSEC in the space of Disse [75], the metastases with a co-option sinusoidal system developed. On contrary, as showed in the other study, if tumoral cells were injected via the portal route, near the portal tracts, they demonstrated the development of an angiogenic pattern [76]. However, at the moment, further investigation is needed to prove this fact. This might explain why the patients with a replacement (presumably non-angiogenic) HGP obtain less clinical benefit from bevacizumab-chemo treatment, which is a VEGF inhibitor, when compared with the patients with desmoplastic (angiogenic) metastases [27]. This concept may explain the failure of anti-angiogenic therapy in breast cancer liver metastases, which predominantly show the replacement HGP [12]. 

Thus, the tumor vascularization strategy used by each HGP is another feature to consider in order to understand their biological processes and finally select the best treatment option in each case.

### 3.4. Immune Infiltrates

The immune system plays crucial role in controlling tumor growth and dissemination. With the introduction of immune checkpoint inhibitors (ICI) into clinical practice, the interest of the research community in the tumor immune microenvironment has been raised and a growing body of preclinical and clinical studies has been accumulated. With this background, the immune microenvironment of LM-CRC has not been fully investigated.

In 2018, Pagès et al. [77] reported a study based on 603 resected synchronous and metachronous metastases from mCRC patients. When compared to primary tumors, metastases were characterized by higher levels of CD3, CD8 and CD45RO lymphocytes and lower levels of CD20 cells, while the FoxP3-positive cell count did not demonstrate a difference. This observation, however, was based on either hot spot analysis or the whole slide average immune cell count and did not consider the spatial context. A certain pattern was observed in relation to the size of the metastasis and lymphocyte infiltration: the average infiltration was higher in small metastases; however, larger-size lesions often had “hot spots” with very high lymphocyte density, not observed in small tumors. This pattern was also seen when analyzing small and large metastases in patients with multiple metastases. Importantly, although the average lymphocyte density did not differ between cases with different metastatic burdens, the individual lesions in patients with multiple metastases demonstrated significant heterogeneity with regard to immune density. This observation is in consistence with earlier work of Halama et al. [78] performed with a focus on partially different immune cell subsets (CD3+, CD8+ and granzyme B+ cells) and overall may suggest that the tumor-related factors are more important for the local immune response than individual variations in the immune background of the host organ. However, Brunner et al. [79], although based on very few samples (*n* = 5), reported a conflicting observation, i.e., the homogeneity of immune infiltration patterns (CD4, CD8, CD45RO) between different lesions in patients with multiple metastases. 

Immune infiltration is not evenly distributed in liver metastases. Thus, peripheral regions of the metastases have higher infiltration of CD3, CD8, C45RO and CD20 lymphocytes [79,80,81,82] and probably of NKp46+ cells [82] in comparison to the central regions of the lesion. Further, peripheral regions have higher infiltration of CD4, CD45RO and CD8 cells in comparison to distant peritumoral liver parenchyma [79]. However, the definition and extent of the peripheral region (or invasive margin) of the metastases are not set up and vary between the studies. For example, sometimes, such peripheral region contained marginal tumor tissue, and other times, it was defined as a peritumoral, non-malignant zone of a certain depth. Attempts for more detailed spatial analysis are rare and until now are limited to very few samples [78].

What is even more important in the contexts of the current review is that the above-mentioned studies did not adjust the scoring technique according to different HGPs and almost never considered the HGP in reports. However, the evaluation of inflammatory infiltrates in the context of HGPs is of crucial importance, since they topographically delimit structures that end up being physical and chemical barriers to infiltration by these cells. Concerning intratumoral regions, FoxP3, CD79A and Kappa/Lambda were more frequent in dHGP according to Höppener et al. [79]. Importantly, only intraepithelial, but not stromal, CD8 cells demonstrated such difference.

A few studies investigated peritumoral tissue. Thus, Brunner et al. reported that high levels of CD4, CD45RO and CD8 in the near peritumoral region (when normalized to immune infiltration in distant peritumoral regions) were associated with the dHGP [79]. This was further developed by Höppener et al. [83], who showed higher counts for CD8, CD45, CD79A, Kappa/Lambda and SLAMF7 cells in peritumoural areas of dHGP CRLMs in comparison to the rHGP. At the same time, FoxP3 cells did not demonstrate such difference. Digital image analysis on an independent tissue set confirmed the results for CD8 cells but also revealed the same association for FoxP3 cells (higher in the peritumoral dHGP). These in situ analyses did not reveal a difference in CD4 infiltration, but flow cytometry demonstrated a relative increase in CD4+ cells within the band of CD3+ T lymphocytes in the non-dHGP. Authors speculated that such effect must be due to an increase in CD4+FoxP3- helper cells because no difference was found for Tregs (CD4+FoxP3+). Interestingly, even in individual metastases with a combination of HGPs, a high density of CD8 cells is seen in the area of the dHGP while being very low in the regions of the rHGP [27].

The concept based on the different spatial patterns of the immune response was developed in recent years. Thus, tumors demonstrating high lymphocyte infiltration were designated “inflamed” or “hot”, while tumors with a low immune cell content were classified as “desert” or “cold” tumors [84]. The third category of tumors was characterized by high abundance of immune cells, which, however, do not penetrate cancer cell nests but are instead retained in the stroma. This phenotype was termed “immune-excluded”. Interestingly, immune infiltration in “immune-excluded” tumors may be limited to the peritumoral stroma or capsule (if present) but may also infiltrate the tumor itself and stay retained in intratumoral stromal bulks [84]. We believe that these two types of “immune-excluded” phenotype may be characterized by qualitative differences in the existing antitumor response and by different mechanisms of the blockage of immune cells. The important observation regarding lymphocyte infiltration in the dHGP of LM-CRC is that the majority of lymphocytes accumulate at the interface between the outer part of the desmoplastic rim and the adjacent liver parenchyma [27]. Most immune cells, thus, although present in a high number, are kept at a distance from the malignant tissue. Due to such an immune phenotype, Jan van Dam [27,85] suggested considering metastases with the dHGP as “immune-excluded” tumors. However, some other reports considered an association of the dHGP with an inflamed phenotype [86]. In the rHGP, lymphocytes are present in the interface between liver parenchyma and the tumor and thus directly approach malignant cells, although being in relatively low numbers. We believe that a certain level of the confusion and discordance in results is caused by several reasons. First, different immune cells and different visualization methods may capture distinct lymphocyte subclasses with potentially different biology. Second, the absence of proper definitions for the immune phenotypes. Third, a lack of standardized criteria for the assessment and reporting of the spatial immune cell distribution in different HGPs.

It remains a question as to why there is such a lack of an immune-competent response from the liver tumor microenvironment [86]. However, different cells can contribute to such unresponsiveness. Thus, cancer-associated fibroblasts can express PD-L1 [87,88,89,90] or may induce its expression in other cells [82,83,84], contributing to immunologic silencing. Particular CAF subtypes can also induce the expression of PD-1 and CTLA4 in surrounding lymphocytes [88,89,90,91], contributing to immunosuppression and resistance to immunotherapy. Liver endothelial sinusoidal cells can also induce T cell exhaustion and suppress innate responses [92]. Kupffer cells [93] and HSC [93] contribute to such hypo-reactivity. However, the balance between responsiveness or unresponsiveness seems to be predetermined according to the HGP.

The overall impact of the prognostic role of immune infiltration was recently summarized elsewhere [94]. Despite heterogeneous study designs, cells of interest and scoring criteria, a clear overall trend with high intratumoral infiltration of CD4, CD8 or CD45RO and improved survival could be stated. Interestingly, out of nine studies, six considered peritumoral regions as of separate interest; however, none of them distinguished HGPs. The prognostic impact of immune cell infiltration in peritumoral regions is not clear. To the best of our knowledge, only one study evaluated both HGPs and immune phenotypes of LM-CRC with regard to survival. In this study, Stremitzer et al. classified 118 metastases into an inflamed immune phenotype and a non-inflamed immune phenotype, based on CD8-positive cell infiltration [86]. The inflamed immune phenotype was associated with the dHGP and both of them were linked to improved survival. Unfortunately, the survival differences between all three groups and their distribution across HGPs remained unreported.

Studies have demonstrated that patients with MSI-H cancer, which develop distant metastases, may benefit from immunotherapy [95,96]. These results led to FDA approval for ICI pembrolizumab and nivolumab for the treatment of chemoresistant MSI-H (microsatellite-instability high) metastatic CRC. Despite pioneering studies of the Galon group [80,97,98], the dynamics of the immune microenvironment under immunotherapy in metastatic lesions remains unclear.

Thus, LM-CRC can have different immune infiltration patterns. The crude rate of lymphocyte immune infiltration in the resected metastatic lesions seems to be expectedly associated with improved survival. However, the role and impact of peritumoral immune infiltration and inflammation status in liver parenchyma and their impact on prognosis are yet to be discovered. There is evidence that certain immune patterns are associated with certain HGPs. The HGP may be one of the key factors sculpting the development of certain immune phenotypes of LM-CRC. Further studies using standardized evaluation and reporting systems are needed to confirm and clarify such association.

## 4. Clinical Opportunities of TME in the Context of the HGP

LM-CRC are a defiance for cancer treatment, since they are entities that have the particularity of recruiting a large number of cells with an immunosuppressive capacity and, even more challenging, that have characteristic resistance to a multitude of antitumor drugs, aspects very well summarized in Ciner AT et al. [99].

However, the particular pathological and histological characteristics of the TME in hepatic metastases, as well as the particular biology of the different phenotypes, are the aspects that could be exploited clinically. We have already commented that HGPs have a prognostic and predictive value, although prospective studies are warranted to validate this. Thus, knowing the HGP prior to surgery, i.e., being able to determine metastases with an established capsule, could contribute to the better selection of patients for a curative surgical resection. Knowing the HGP at the pre-operative stage could improve patient selection for neoadjuvant treatment, minimizing the toxicity effects and surgery-related risks. This fact could be approached by using modern medical imaging techniques and is extensively reviewed by Oliveira RC et al. and Latacz E et al. [21,85]. Radiomics is another powerful non-invasive tool capable of classifying tumors at the morphological [100] and molecular levels [101]. Thus, in a retrospective study, Cheng et al. used pre- and post-contrast (arterial and portal venous) phase MDCT images to build a radiomic classifier which predicted the HGP with remarkable accuracy [102]. According to the authors, they could distinguish a clear tumor–liver interface in the tumors with a dHGP and fibrotic capsule, while a poorly defined tumor–liver interface would be associated with a replacement growth pattern.

Another important feature of liver metastases, linked to the HGP, is the type of vascularization which the tumor develops. As we described above, two vascularization strategies, i.e., vessel co-option (in tumors with a replacement HGP) or neoangiogenesis (in tumors with a dHGP), can be clearly distinguished. In this regard, patients whose metastases have a desmoplastic HGP would be candidates for receiving anti-angiogenics drugs in neoadjuvant settings. Non-invasive imaging for the classification of the HGP is the pre-requisite for such therapeutic approach.

Another opportunity relies on the observation that regardless of the HGP, LM-CRC are characterized by some degree of immune suppression, at least in central areas of the tumor lesion, as above reviewed. However, anti-angiogenic [103] and anti-fibrotic [104,105,106,107] therapies could be interesting strategies to re-establish the anti-tumoral immunity and are thus direct targets of immunotherapeutic approaches. Current clinical trials are assessing this hypothesis (NCT03698461), although no criteria for selection of patients according to the HGP have been applied. However, some caution must be exercised since some approaches have turned out to be counterproductive and report the opposite observation [108], especially if they are not accompanied by ICI. 

The counterproductive results obtained so far using the deletion or silencing of particular mesenchymal cells raise the question for different CAF subsets, as stated throughout the text. This fact is especially relevant in the context of HGPs and taking into account different potential precursors of CAFs in the liver. It could be hypothesized that the fibroblasts that form the capsule are different from those that are located in the central regions of the tumor. Thus, the fibroblasts that form the fibrotic rim may exert a protective role by enveloping the tumor and hindering the invasion into the liver tissue. The capsule in the dHGP may have certain similarities to the peritumoral host reaction against benign tumors or abscesses. 

Thus, these observations open the door for researchers in the field to try to reprogram intratumoral fibroblasts. We usually find these cells in large numbers in rHGP metastases but also in dHGP metastases. However, intratumoral CAFs in invasive versus central deeper areas could probably be performing different functions. The conceptual idea is to differentiate those CAFs into fibroblasts that would have the ability to envelop, as a shield or shell, the tumor itself, preventing the spreading of the tumor inside the liver plates. Reprogramming fibroblasts has been a successful strategy at the preclinical level so far. As in all organs, resident mesenchymal cells are transformed into CAFs due to the selective pressures exerted by the tumor–stroma crosstalk. In pancreatic cancer, as well as in pancreatitis, pancreatic stellate cells (PSC) are differentiated to CAFs. In this process of loss of quiescence, they lose the expression of the vitamin D receptor, a fact that has been reversed by administering synthetic derivatives of vitamin D such as calcipotriol, inducing quiescence in the CAFs, and, in turn, the reestablishment of the transcriptional programs of the PSC [109]. In the same vein, all-trans retinoic acid (ATRA), the active metabolite of vitamin A, is able to restore quiescence of PSC by means of a reduction in the myofibroblastic properties of activated PSC, a process involving RARβ [110]. However, it is still a matter of major concert determining which is the trigger that induces the encapsulating response. Is it a host response or tumor-driven?

Thus, the particular histology of colorectal carcinoma liver metastases, conditioned by the characteristics of the TME, should be taken into account when designing therapeutic clinical trials and could be known in advance using conventional medical imaging techniques.

## 5. Conclusions

Thus, as final thoughts and comments, LM-CRC are tumors that have particular characteristics in terms of their content in TME cells, both qualitatively and quantitatively. This fact makes them distinct entities with respect to their corresponding primary tumors. Furthermore, this fact should have an impact on the way we treat metastases. However, treatments have not been designed according to these particularities so far. Be that as it may, we need to better understand the particularities of the cells that form the TME in each particular location, since we already know that, e.g., fibroblasts display anatomic demarcation that affects their transcriptional repertoire. 

All these facts acquire even greater relevance since LM-CRC present different HGPs, having a great impact on the way the stroma is organized. In this sense, the new spatial transcriptomic technologies can contribute to improving such knowledge. However, much more affordable and feasible technologies could also contribute with relevant clinical answers. The prospective assessment of the HGP is needed in terms of reinforcing the prognostic and predictive value described so far. Further, it would be highly recommended to include such feasible assessment in the pathological anatomy reports on a routine basis. Furthermore, having this information available prior to surgery could, on the one hand, help design more efficient neoadjuvant treatments for those resectable patients and, on the other hand, be able to offer potentially curative strategies to those initially non-surgical patients. This clinical information can only be obtained through medical imaging techniques, and there are different initiatives underway to demonstrate the feasibility of this option. However, in order to obtain a clinical benefit with an impact on affected patients, it is necessary to deepen the knowledge of the biology of HGPs. We still do not know with certainty which is the cause that motivates the formation of the capsules. However, it would be of great interest to be able to pharmacologically induce the formation of such fibrotic capsules. In any case, the combination of both better knowledge and routine assessment by means of medical imaging would have an impact on the selection of patients for personalized treatments, particularly relevant for the majority of patients that could not benefit from a potentially curative resection.

## Figures and Tables

**Figure 1 ijms-22-01544-f001:**
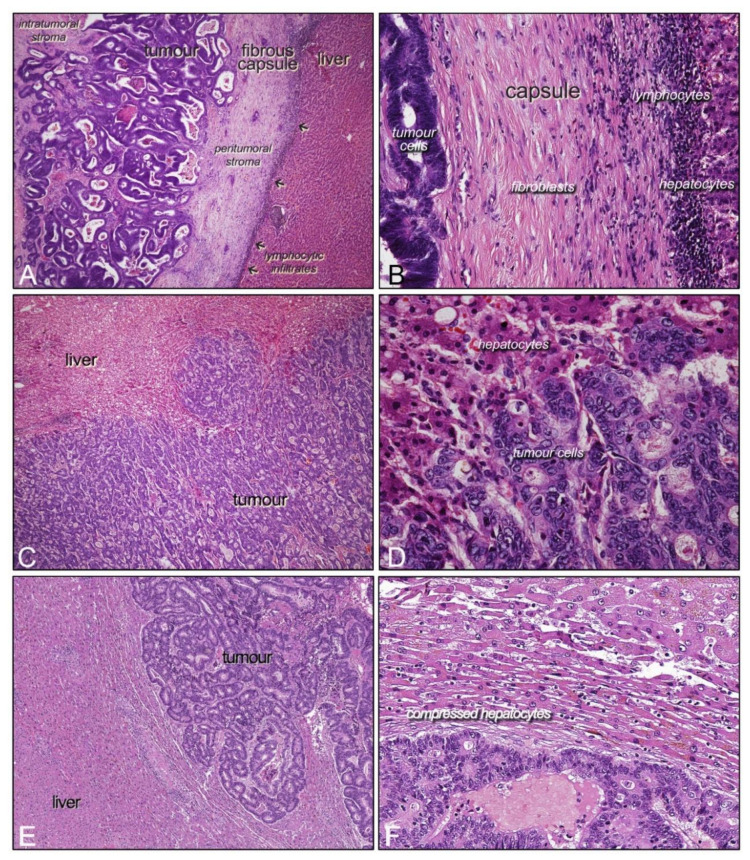
Hematoxylin and eosin staining of histologic growth patterns (HGPs) of liver metastases from colorectal cancer (LM-CRC). (**A**,**B**) encapsulating/desmoplastic, (**C**,**D**) invasive/replacement, and (**E**,**F**) expansive/pushing, the different HGPs at low magnification. In the right column, same patterns at detailed magnification.

**Figure 2 ijms-22-01544-f002:**
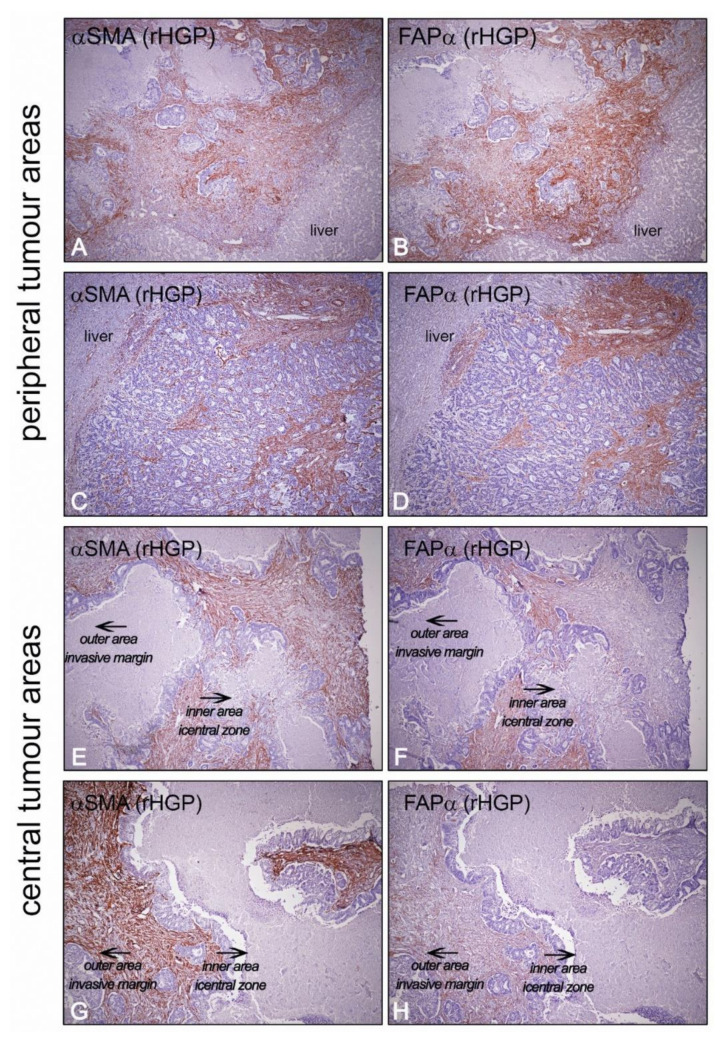
Different cancer-associated fibroblast (CAF) subsets might be topographically located in particular areas inside the LM-CRC. Although it is a fact that should be explored in detail with dedicated technologies such as spatial transcriptomics, invasive margins (**A**–**D**) displayed an intense staining of both αSMA and FAP. The outer regions of the liver metastases are those areas where there is a constant tissue remodeling and where tumor cells show more aggressive phenotypes, higher proliferation and migration capabilities. On the contrary, in central areas of the tumor (**E**–**H**), having the liver–tumor interface as a reference, a progressive decrease in FAP staining is observed, becoming negative in the most central and internal areas of the tumor. The staining of αSMA is quite homogeneous. These areas usually correspond to areas of secondary fibrosis enriched in myofibroblasts, probably corresponding to myCAFs. These areas are progressively repaired as the tumor grows. In these areas, the activation pattern of CAFs is different from those closest to the invasive margin (rHGP stands for the replacement histologic growth pattern).

**Figure 3 ijms-22-01544-f003:**
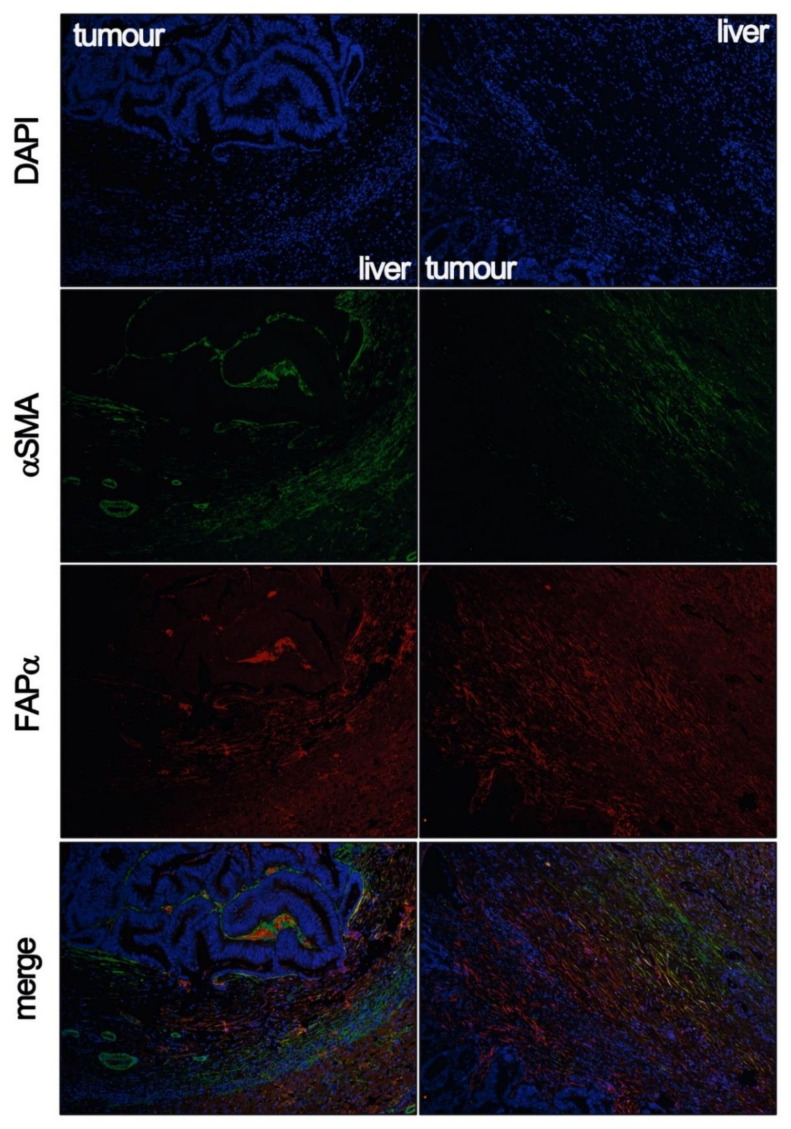
The fibrous capsules of liver metastases present CAFs that express different biomarkers classically associated with the activation of these cells. This fact enables two hypotheses, pending to be proved. On the one hand, it could be that different subpopulations coexisted in the same peritumoral area. On the other hand, it could be the same cells, which, depending on their proximity to the tumor cells, could establish a paracrine crosstalk with them, which would modulate the phenotype of the CAFs. The two columns of images correspond to two different encapsulating tumors, where positive αSMA CAFs (green immunofluorescence) are observed in the outer area of the capsule, the one closest to the liver parenchyma. On the contrary, in the inner area, αSMA positivity progressively disappears, while FAP-expressing cells (red immunofluorescence) appear as fibroblasts approach tumor cells. The FAP protein has two main locations, as a protein anchored to the plasma membrane, or as a component of the extracellular matrix (ECM). In some publications, it has been mentioned that fibrous capsules are formed by fibroblasts on their outermost layer and by acellular components on their innermost layer, mainly ECM components. However, DAPI staining indicates the presence of elongated nuclei that are compatible with the existence of CAFs in the deeper areas of the fibrous capsule.

**Figure 4 ijms-22-01544-f004:**
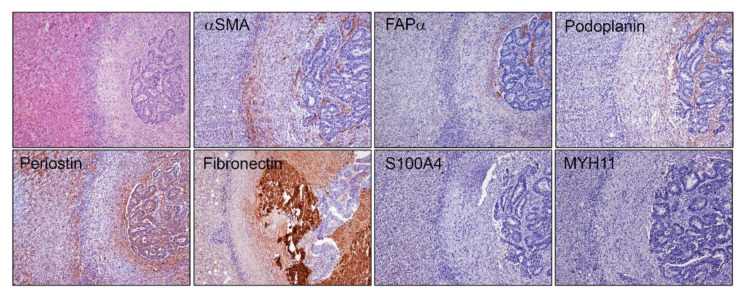
Immunohistochemical staining of a desmoplastic liver metastasis from CRC. We used classical markers for carcinoma-associated fibroblasts (αSMA, FAPα and S100A4). As shown in the microphotographs, the outer half of the capsule stains for αSMA-positive fibroblastic cells, while the inner half does it for FAPα, as also shown in Figure 3. We did not observe staining for S100A4 (also known as FSP1, Fibroblast Specific Protein 1) on cells of the capsule. Podoplanin, which is a type 1 integral membrane glycoprotein and is characteristic of particular CAF subsets, also stains in cells of the inner half of the membrane, as is the case for various proteins of the extracellular matrix, such as fibronectin and periostin (among others, not shown).

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
