# Peer review of "The Tumor Microenvironment in Liver Metastases from Colorectal Carcinoma in the Context of the Histologic Growth Patterns"

_ijms, 2021, doi:10.3390/ijms22041544_

Round 1
Reviewer 1 Report
1) Suggested reference
The four phases of the hepatic metastasis process (see section on BIOLOGY OF THE LM-CRC) were proposed for first time by Vidal-Vanaclocha in 2011:
Vidal-Vanaclocha F. The Tumor microenvironment at different stages of hepatic metastasis. In: Brodt P, editor. Liver metastasis: biology and clinical management. 1st ed. Dordecht (Netherlands): Springer; 2011. p. 43–87.
2) New reference
Ten years earlier than ref 30-37-38, Solaun et al already reported that LSEC is a key actor for HGP development as they are closely related to the metastasis vascularization pattern.
Consider below reference:
Solaun MS, Mendoza L, De Luca M, Gutierrez V, Lopez M-P, Olaso E, Sim BKL, Vidal-Vanaclocha F (2002) Endostatin inhibits murine colon carcinoma sinusoidal-type metastases by preferential targeting of hepatic sinusoidal endothelium. Hepatology 35: 1104–1116
Suggested rewriting for "In addition, LSEC can also be a key for the HGP developing as they as closely related with the kind of metastasis vascularization (30,37,38)."
In addition, LSECs can also be a key player for HGP development as they are closely related to the metastasis vascularization pattern (30,37,38).
3) Suggested rewriting and new reference
The possibility that histopathological growth patterns help as biomarker for immunomodulatory therapy is a relevant concept. However, below paragraph is confusing:
"Finally, LSECs are endocytic cells that scavenge molecules form the bloodstream and present them to hepatocytes. In order to avoid immune responses, that would be unnecessary in physiological conditions, they push immune regulation towards tolerance rather than immunity. Thus, these cells could be mediating the immune suppression that appears specially in replacement HGP (27)."See proposed changes and suggested reference:
Finally, LSECs are endocytic cells that scavenge molecules from the bloodstream and present them to hepatic lymphocytes, which leads to a physiological immune regulation towards tolerance preventing liver parenchymal damage. However, during hepatic metastasis development, tumor-activated LSECs become highly inflamed which foster their immune suppressive effect, specially in replacement HGP (27). As shown in an experimental model of hepatic CRC metastasis, the mechanism is contributed by IL-1-induced LSEC mannose receptors (Arteta et al., 2010)
Suggested reference:
Arteta B, Lasuen N, Lopategi A, Sveinbjörnsson B, Smedsrød B, Vidal-Vanaclocha F. Colon carcinoma cell interaction with liver sinusoidal endothelium inhibits organ-specific antitumor immunity through interleukin-1-induced mannose receptor in mice. Hepatology. 2010 Jun;51(6):2172-82. doi: 10.1002/hep.23590. PMID: 20513002.
4) Proposed rewriting for "They are also implied in endothelial cells recruitment and neovascularization, as well as tumor cell invasion and proliferation (33)":
They are also involved in endothelial cell recruitment and neoangiogenesis (Olaso et al, 2003), as well as in tumor cell invasion and proliferation (33).
and suggested ref.:
Olaso E, Salado C, Egilegor E, Gutierrez V, Santisteban A, Sancho-Bru P, Friedman SL, Vidal-Vanaclocha F. Proangiogenic role of tumor-activated hepatic stellate cells in experimental melanoma metastasis. Hepatology. 2003 Mar;37(3):674-85. doi: 10.1053/jhep.2003.50068. PMID: 12601365.
5) Proposed rewriting and suggested reference for "Therefore, all this knowledge gives leads to speculate that each HGP presents CAFs coming from a main origin of the described above. This would be in line with the hypothesis of the route of access of the tumoral cells to hepatic tissue."
Therefore, all this knowledge leads us to speculate that each HGP contains CAFs coming from a distinct origin among those earlier described. This would be in line with the hypothesis on the HGP predetermination by the different access routes of circulating tumor cells into hepatic tissue (Vidal-Vanaclocha 2011).
6) Authors should spell out MSI-H
7) Proposed rewriting for: "Another opportunity relies on the observation, that regardless of the HGP, LM-CRC are characterized relative immune suppression, at least in the intratumoral central areas of the lesion, as extensively reviewed above. "
Another opportunity relies on the observation that regardless of the HGP, LM-CRC are characterized by some degree of immune suppression, at least in central areas of the tumor lesion, as above reviewed.
Author Response
As suggested by the reviewer we have rewritten some paragraphs having into account tha changes and references suggested by the reviewer #1.
We hope that changes made make statements clearer for the reader.
The new text is highlighted in yellow in the new document.
Reviewer 2 Report
Investigations on tumor microenvironment in colo-rectal liver metastases combined with descriptions of the histologic growth patterns are certainly important and potentially useful in future treatment strategies but after more than ten years of laboratory research clinical state of the art in treatment and follow up of patients with colo-rectal liver metastases does not consider such characteristics. And why not? Firstly, we do not have sensitive imaging methodologies – it be CT scan, PET-CT, MRI or ultrasound – yielding imaging data which reliably reflects the microscopically proven growth patterns. Liver biopsies may be helpful but must be avoided in patients in whom subsequent liver surgery (or another ablative intervention) is planned. Secondly, the microscopic growth pattern of a removed liver metastasis may have prognostic impact but the magnitude of the influence is weak compared to other - clinical - prognostic factors why HGP have not been enrolled into current follow up programs. Immunotherapy has proven very efficient in MSI-H cancers and in cases with definitively unresectable liver metastases it would be fully ethically acceptable to obtain biopsies prior to initiation of treatment. If conventional chemotherapy is planned as first line before subsequent pembrolizumab biopsies would be warranted both before and after the chemotherapy in order to relate HGP characteristics and changes on chemo to the effect of the immunotherapy.
The paper reads well, it is exhaustive and well balanced and gives a useful up to date overview of laboratory research in the field. But it would improve the paper if the authors tried to conclude on where we are now and where to go (and how) hand in hand with the clinicians and the pharmaceutical industry.
Author Response
In agreement with the reviewer, we also consider that a final section of conclusions and future directions will make a better ending message for the reader. So, we added a "conclusions" section as it follows...
CONCLUSIONS
Thus, as final thoughts and comments, LM-CRC are tumors that have particular characteristics in terms of their content in TME cells, both qualitatively and quantitatively. This fact makes them distinct entities with respect to their corresponding primary tumors. And this fact should have an impact on the way we treat metastases. However, treatments are not designed according to these particularities so far. Be that as it may, we need to better understand the particularities of the cells that form the TME in each particular location, since we already know that e.g. fibroblasts display anatomic demarcation that affects their transcriptional repertoire.
All these facts acquire even greater relevance since LM-CRC present different HGP, having a great impact on the way how the stroma is organized. In this sense, the new spatial transcriptomic technologies can contribute to improving such knowledge. However, much more affordable and feasible technologies could also contribute with relevant clinical answers. The prospective assessment of the HGP is needed in terms of reinforcing the prognostic and predictive value described so far. And it would be highly recommended to include such feasible assessment in the pathological anatomy reports on a routine basis. Furthermore, having this information available prior to surgery could, on the one hand, design more efficient neoadjuvant treatments for those resectable patients, and secondly, be able to offer potentially curative strategies to those initially non-surgical patients. This clinical information can only be obtained through medical imaging techniques, and there are different initiatives underway to demonstrate the feasibility of this option. However, in order to obtain a clinical benefit with an impact on affected patients, it is necessary to deepen the knowledge of the biology of HGP. We still do not know with certainty which is the cause that motivates the formation of the capsules. However, it would be of great interest to be able to pharmacologically induce the formation of such fibrotic capsules. In any case, the combination of both, better knowledge and routine assessment by means of medical imaging would have an impact on the selection of patients for personalized treatments, particularly relevant for the majority of patients that could not benefit from a potentially curative resection.